# Identification of shared bacterial strains in the vaginal microbiota of related and unrelated reproductive-age mothers and daughters using genome-resolved metagenomics

**Michael T. France**[1,2], **Sarah E. Brown**[3], **Anne M. Rompalo**[4], **Rebecca M. Brotman**[1,3], **Jacques Ravel**[1,2]*

**1** Institute for Genome Sciences, University of Maryland School of Medicine, Baltimore, Maryland, United States of America, **2** Department of Microbiology and Immunology, University of Maryland School of Medicine, Baltimore, Maryland, United States of America, **3** Department of Epidemiology and Public Health, University of Maryland School of Medicine, Baltimore, Maryland, United States of America, **4** Division of Infectious Diseases, John Hopkins School of Medicine, Baltimore, Maryland, United States of America

* jravel@som.umaryland.edu

**Data Availability Statement:** Raw 16S rRNA amplicon and human removed shotgun metagenomic sequence data has been deposited in

## Abstract

It has been suggested that the human microbiome might be vertically transmitted from mother to offspring and that early colonizers may play a critical role in development of the immune system. Studies have shown limited support for the vertical transmission of the intestinal microbiota but the derivation of the vaginal microbiota remains largely unknown. Although the vaginal microbiota of children and reproductive age women differ in composition, the vaginal microbiota could be vertically transmitted. To determine whether there was any support for this hypothesis, we examined the vaginal microbiota of daughter-mother pairs from the Baltimore metropolitan area (ages 14–27, 32–51; n = 39). We assessed whether the daughter's microbiota was similar in composition to their mother's using meta-taxonomics. Permutation tests revealed that while some pairs did have similar vaginal microbiota, the degree of similarity did not exceed that expected by chance. Genome-resolved metagenomics was used to identify shared bacterial strains in a subset of the families (n = 22). We found a small number of bacterial strains that were shared between mother-daughter pairs but identified more shared strains between individuals from different families, indicating that vaginal bacteria may display biogeographic patterns. Earlier-in-life studies are needed to demonstrate vertical transmission of the vaginal microbiota.

## Introduction

The human body is colonized by microbial populations which together comprise our microbiota [1]. These populations have been shown to be critical determinants of our health and well-being [2,3] and are founded early in life [4]. Initial colonization of newborn infants primarily occurs during and immediately following birth [5,6], although there is an active debate on whether *in utero* seeding plays a role in this early colonization [7–13]. Establishment of the

the NCBI SRA under the Bioproject PRJNA779415 (https://www.ncbi.nlm.nih.gov/bioproject/ PRJNA779415). Python and R scripts used in the analysis of these data and in the generation of figures are available in the Github database (https:// github.com/ravel-lab/MotherDaughter).

**Funding:** This work was supported in part by funding from the National Institute of Child Health and Development (https://www.nichd.nih.gov/), the National Institute of Allergy and Infectious Disease (https://www.niaid.nih.gov/), and the National Institute of Nursing Research (https://www.ninr. nih.gov/) under the grants R01AI060892 (AR), R01NR015495 (JR), and R01AI116799(RB). The funders had no role in study design, data collection and analysis, decision to publish, or preparation of the manuscript.

**Competing interests:** Dr. Ravel is a co-founder of LUCA Biologics, a biotechnology company focused on translating microbiome research into live biotherapeutics for women's health. This does not alter our adherence to PLOS ONE policies on sharing data and materials. All other authors declare that they have no competing interests.

microbiota is theorized to be critical to the programming of the neonatal immune system [14–17]. The maternal microbiota has long been hypothesized to be a major contributor of microbial strains to their newborn offspring through a process of vertical transmission [18]. As the neonate moves through the vaginal canal, it is expected to be exposed to the mother's vaginal microbiota and perhaps their fecal and skin microbiota. It follows then that the microbiota of neonates born via C-section has been observed to transiently differ in composition from those born via vaginal delivery [14,19,20]. Studies seeking to demonstrate this process of vertical transmission have identified shared bacterial phylotypes in the microbiota of mothers and their neonates using 16S rRNA amplicon sequencing [21,22]. However, these data lack the resolution necessary to identify shared strains [23]. The most convincing evidence for vertical transmission comes from studies which either used cultivation or shotgun metagenomic based techniques to identify bacterial strains in the microbiota of mothers and their infants [24–28]. Yet, these probable vertically transmitted strains have been shown to be minority members of the neonate's microbiota and to be short-lived [24]. More study is needed to define the provenance of a neonate's microbiota.

Much of the work on maternal microbiota transmission has focused on the neonate's intestinal, skin, or oral microbiota [24–28]. The source of the bacterial species and strains that inhabit the vagina is not known. Reproductive-age women routinely have communities which are dominated by *Lactobacillus* with *L. crispatus*, *L. iners*, *L. jensenii*, and *L. gasseri* being the most prevalent species [29,30]. A significant proportion of these women, however, have communities which do not contain a high relative abundance of lactobacilli and instead are characterized by a more even distribution of several obligate or facultative anaerobes including species in the *Gardnerella*, *Atopobium*, and *Prevotella* genera [29,30]. These *Lactobacillus* deficient communities are more common among women of Hispanic and African descent [29–31] and have been associated with increased risk for adverse health events, including reproductive tract infections [32–36]. Less is known about the microbial communities which comprise the vaginal microbiota of pre-pubertal children, but they have been shown to differ in composition and in bacterial load from those found in reproductive-age women [37–39]. Any bacteria which are transferred at the time of birth may not be capable of surviving in a child's vagina, which has been shown to have neutral or alkaline pH and to have a paucity of *Lactobacillus* [38]. It is not until early puberty that the species which are common in a reproductive age women's vaginal microbiota (e.g. *L. crispatus*, *L. iners*, *G. vaginalis*) gain dominance in an adolescent's vaginal microbiota [40]. While it is thought that pubertal hormonal and physiological changes which occur are responsible for this shift in the composition of the vaginal microbiota [41–43], it is not clear where the strains come from. It could be that they are of maternal origin and have persisted at low abundance throughout early-life or that they are acquired later in life through some other mechanism.

To determine whether there was evidence for the vertical transmission of the vaginal microbiota which persisted into adolescence and adulthood, we characterized the vaginal microbiota of pre-menopausal mothers and their post-menarcheal daughters. Metataxonomics was used to investigate similarities in community composition between the mother-daughter pairs and genome-resolved metagenomics was used to identify bacterial strains which the two had in common.

## Methods

### Cohort description/sample collection

We characterized the vaginal microbiota of 87 reproductive-age women including 45 daughters and their 42 mothers. The average age of the daughters was 19 (14–35) and the average age

of the mothers was 41 (32–51). Participants included in this study self-identified as Black or African American and took part in a douching intervention study [44]. Data was not collected on the mode of birth (vaginal delivery versus c-section), although the estimated rate in the US around the time of the daughter's births was around 22% [45]. Vaginal swab specimens were collected and stored at -80˚C. All participants or their legal guardians (in case of minors) provided written informed consent prior to enrollment in the study. In addition, all minor participants provided written assent to their participation in the study. All procedures were conducted in accordance with relevant guidelines and regulations and was approved by the internal review boards at the University of Maryland Baltimore (#HP-00045398) and the Johns Hopkins University School of Medicine (#NA-00004835).

## DNA extraction

DNA was extracted from 200 μL of vaginal swab specimen resuspended in 1ml of phosphate buffered saline transport medium. DNA extractions were performed using the MagAttract PowerMicrobiome DNA/RNA Kit (Qiagen; Hilden, Germany) and bead-beating on a Tissue-Lyser II (Qiagen) according to the manufacturer's instructions and automated onto a Hamilton STAR robotic platform (Hamilton Robotics; Reno, NV, USA).

## 16S rRNA gene sequencing

The V3V4 region of the 16S rRNA gene was amplified and sequenced as described previously [46]. The protocol utilizes two amplification steps: one which targets the V3V4 region and one which adds barcoded sequencing (primer sequences in S1 Table). Pooled amplicons were then sequenced an Illumina HiSeq 2500 (Illumina; San Diego, CA, USA) and the resulting paired end sequence reads were processed using DADA2 [47] to identify amplicon sequence variants (ASVs) and remove chimeric sequences as described previously [29]. The median number of sequences per sample following processing was 16 990 (range: 182–36 401). Each ASV was assigned to a taxonomic group using the RDP Naïve Bayesian Classifier [48] trained with the SILVA 16S rRNA gene database [49]. Genera common in the vaginal environment (e.g. *Lactobacillus*, *Gardnerella*, *Prevotella*, *Sneathia*, and *Mobiluncus*) were further classified at the species level using speciateIT (version 1.0, http://ravel-lab.org/speciateIT). The sequence counts attributed to ASVs assigned to the same phylotype were added together. Samples for which less than 500 sequences were generated were dropped from the analysis. Phylotypes with a study-wide average relative abundance of $< 10^{-4}$ were removed. The final dataset contained 81 samples and 100 phylotypes and can be found in S1 Table. Taxonomic profiles were assigned to community state types (CSTs) using VALENCIA [29]. Similarity in taxonomic composition between mother-daughter pairs was assessed using the Yue-Clayton θ [50]. A permutation test was performed to determine whether these observed values differed with that expected by chance alone. Taxonomic profiles for mothers and daughters were each shuffled randomly and then similarity was assessed in the same manner. This process was repeated 100 times.

## Shotgun metagenomics

Shotgun metagenomic data was generated for 22 families and included 47 total samples (22 mothers and 25 daughters). The samples selected for shotgun metagenomics are indicated in Fig 1. Families which either had similar taxonomic profiles or had species in common were selected for this analysis. Shotgun metagenomic sequence libraries were prepared from the extracted DNA using Illumina Nextera XT Flex kits according to manufacturer recommendations. The resulting libraries were sequenced on an Illumina HiSeq 4000 (10 per lane,150 bp paired-end mode) at the Genomic Resource Center at the University of Maryland School of

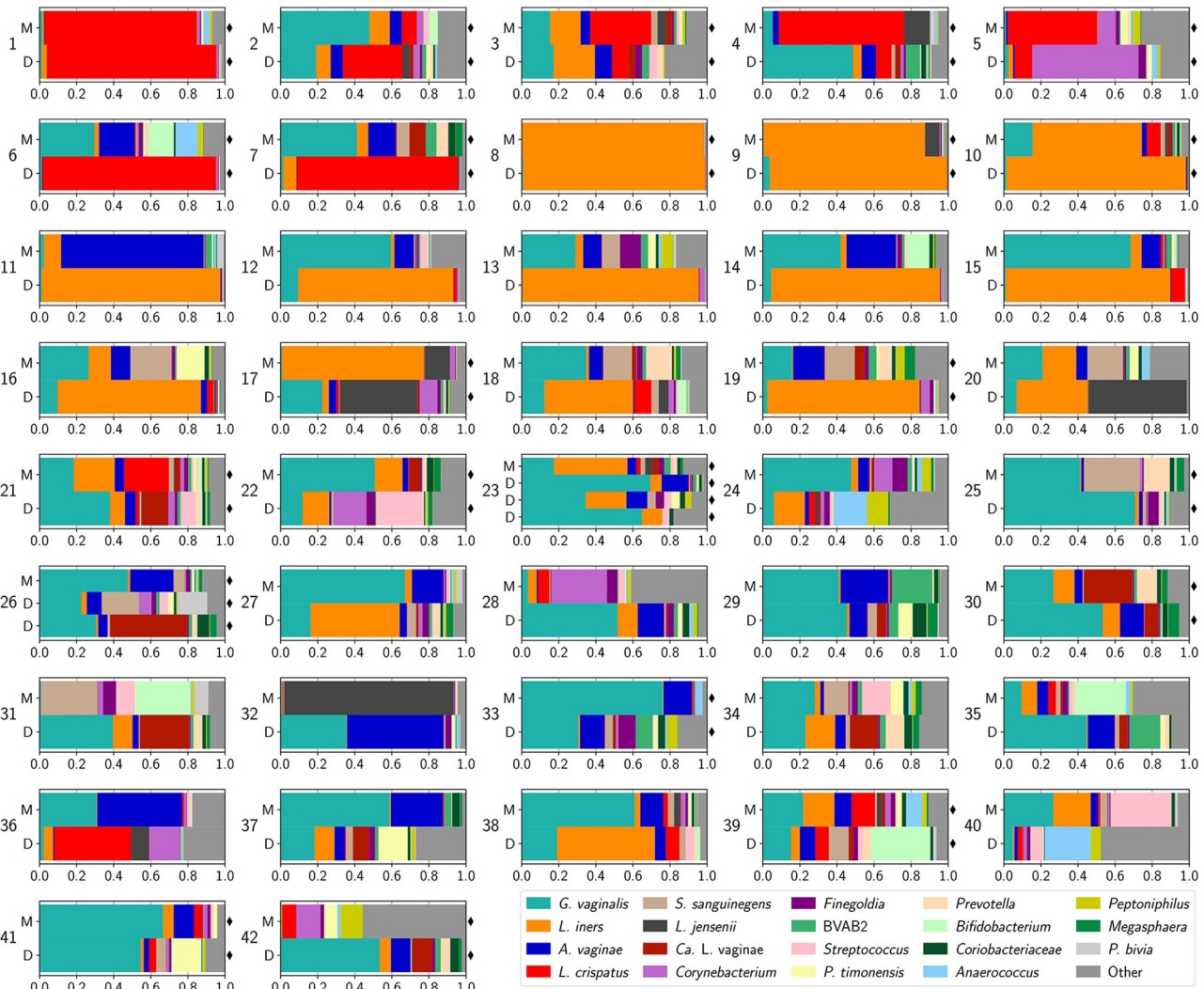

**Fig 1. Taxonomic composition of the vaginal microbiota of reproductive age mothers and daughters.** Stacked bars represent the relative abundances of individual bacterial phylotypes. Each plot displays the profiles for members belonging to the same family (M-Mother, D-Daughter). Two families (23,27) had 2 and 3 daughters, respectively. Samples denoted with black diamonds were selected for shotgun metagenomic analysis.

Medicine. The average number of read pairs generated for each library was 37,000,000 (range: 24,500,000 to 81,800,000). Human reads were identified in the resulting sequence datasets using BMtagger and removed (ftp://ftp.ncbi.nlm.nih.gov/pub/agarwala/bmtagger/). Sequence datasets were further processed using sortmeRNA [51] to identify and remove ribosomal RNA reads and the remaining reads were trimmed for quality (4 bp sliding window, average quality score threshold Q15) using Trimmomatic v0.3653 [52]. Reads trimmed to less than 75bp were removed from the dataset.

## Genome resolved metagenomics

The taxonomic composition of each metagenome was established by mapping to the VIRGO non-redundant gene catalog [53]. *De novo* assembly was performed on each metagenome using metaspades [54,55] with k-mer sizes: 21, 33, 55, 77, 99, 101, and 127. The resulting assemblies were separated into single genome bins using a reference guided approach. For

each metagenome, the sequence reads were mapped back to the corresponding assembly, to establish the contig coverage, and to the VIRGO gene catalog, to establish taxonomy of the contig. Contigs demonstrating at least 5X coverage and which were found to have at least 90% of the reads mapping to VIRGO genes with the same taxonomic annotation were separated into species bins. The species bins were further split into metagenome assembled genomes (MAGs) based on differences in contig coverage. Quality of the resulting MAGs were examined using checkM [56] and those demonstrating at least 80% completion and less than 5% contamination were used in the subsequent analyses (S2 Table). The average completeness of the MAGs was 97.04% (80.9%-100%) and the average contamination was 1.05% (0.0%-4.94%). Genes were identified in each MAG using prodigal [57] and OrthoMCL was used to identify those which were common to at least 95% of the MAGs [58]. Thirteen such genes were identified and their amino acid sequences were individually aligned using Muscle [59] and then concatenated into a single alignment using phyutility [60]. PartitionFinder was used to select an appropriate partitioning scheme and model of molecular evolution [61]. The Phylogeny of the 225 MAGs was established using RaxML-ng with 10 parsimony and ten random starting trees [62]. Bootstrap convergence was detected using the autoMRE setting and occurred after 750 replicates. Relative abundance of MAGs in their resident communities was approximated as the percent of reads from the metagenome mapping to the MAG.

### Identification of shared strains

Similarity between the MAGs was assessed using inStrain [63]. An all-versus-all strategy was used wherein a separate inStrain profile was built by mapping the sequence reads from each metagenome against the MAGs recovered from each participant using Bowtie2 [64]. This resulted in 2209 inStrain profiles. For each participant, the set of 47 inStrain profiles were then summarized using the inStrain compare function with Ward linkage, which afforded the determination of coverage overlap and sequence similarity between the sequences reads of each metagenome and the MAGs of each participant. We then applied a stringent sequence similarity threshold (70% coverage and at least 99.9% sequence similarity) to identify participants with shared bacterial strains. A network diagram representing strain sharing was built from the using the NetworkX python package (https://networkx.org/). Sequence similarity between shared strains identified in the same family versus different families were compared using a Wilcoxon rank sum test.

## Results

### Similarity in the taxonomic composition of mothers and their daughters

We first asked whether mothers and their daughters had vaginal microbiota which were similar in taxonomic composition. Metataxonomics was used to assess the composition of the vaginal microbiota for 42 mothers and their 45 daughters (Fig 1). One family had two daughters, and another had three (26 & 23, respectively). Similarity between communities was assessed using Yue & Clayton's θ, which is function of the relative abundances of shared and non-shared species in the communities. While there were several examples of mother-daughter pairs which had very similar in taxonomic composition (e.g. families 1, 8, 23), there were also several which did not (e.g. families 6, 15, 24). The average similarity between mother-daughter pairs was 0.3, indicating that most were found to have communities with different compositions. Similarity between mother-daughter pairs was higher when the daughter had a community state type (CST) that was not dominated by *Lactobacillus* (CST IV, Fig 2A).

Because the taxonomic composition of the human vaginal microbiota routinely resembles one of a limited number of configurations, it is expected that there can be a degree of similarity

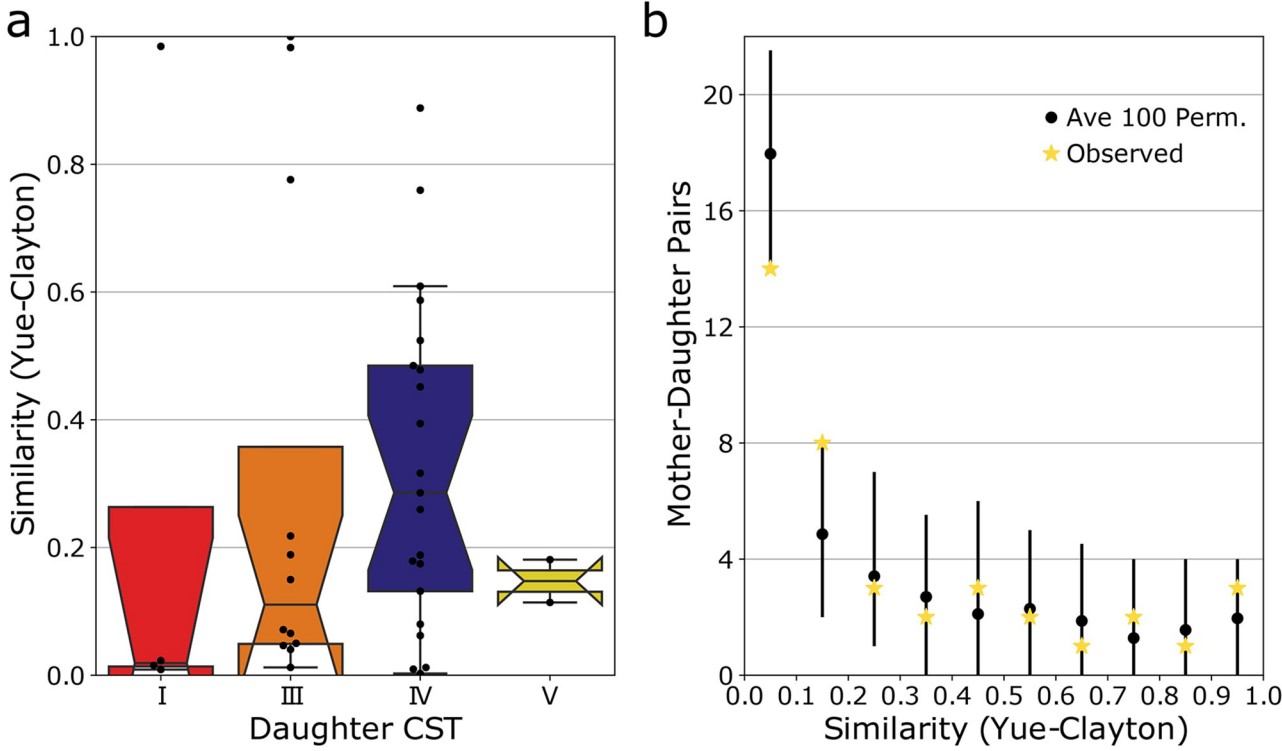

**Fig 2.** Similarity in the taxonomic composition between mother-daughter pairs, delineated by the daughter's CST assignment (A). Higher values of the Yue-Clayton index signify communities that bear greater taxonomic compositional similarity. Permutation tests were used to establish whether the observed similarities between mothers and their daughters were different than that expected by chance alone (B). Black points represent the average number of permuted mother-daughter pairs whose similarity fell within 0.1 increments of the Yue-Clayton similarity index, while yellow stars represent the observed number of pairs. Error bars span the range between the 2.5% and 97.5% quantiles of the 100 random permutations.

between entirely unrelated individuals. To determine whether the observed similarity between mother-daughter pairs exceeded that expected by chance alone we used a permutation test. Taxonomic profiles were shuffled and the similarity between these randomized mother-daughter pairs was assessed in the same manner. As can be seen in Fig 2B, the distribution of similarity scores for the permuted data did not differ substantially from the observed distribution. The observed data was found to have slightly fewer pairs with θ value between 0–0.1 (p = 0.02); and slightly more pairs with a θ value between 0.1–0.2 (p = 0.02). This result indicates that the observed similarity in composition between mother-daughter pairs was not different than that for randomly selected mother-daughter pairs.

## Genome resolved metagenomics

In the above analysis we demonstrated that most mother-daughter pairs did not have vaginal microbiota with similar taxonomic profiles. Yet, many pairs were found to have species in common, just at different relative abundances (e.g. *L. crispatus* in family 4). To determine whether the populations of these shared species were comprised of the same strain(s), we selected 22 families (47 participants) to conduct shotgun metagenomic sequencing (denoted by black diamonds in Fig 1). The resulting metagenomes were assembled and binned allowing us to recover 225 near-complete MAGs (Fig 3). We recovered about 5 MAGs per metagenome with a minimum of 1 and a maximum of 13. The recovered MAGs included: 14 *Atopobium*, 12 "*Ca*. Lachnocurva vaginae", 53 *Gardnerella*, 16 *L. crispatus*, 27 *L. iners*, 6 *L. jensenii*, 29

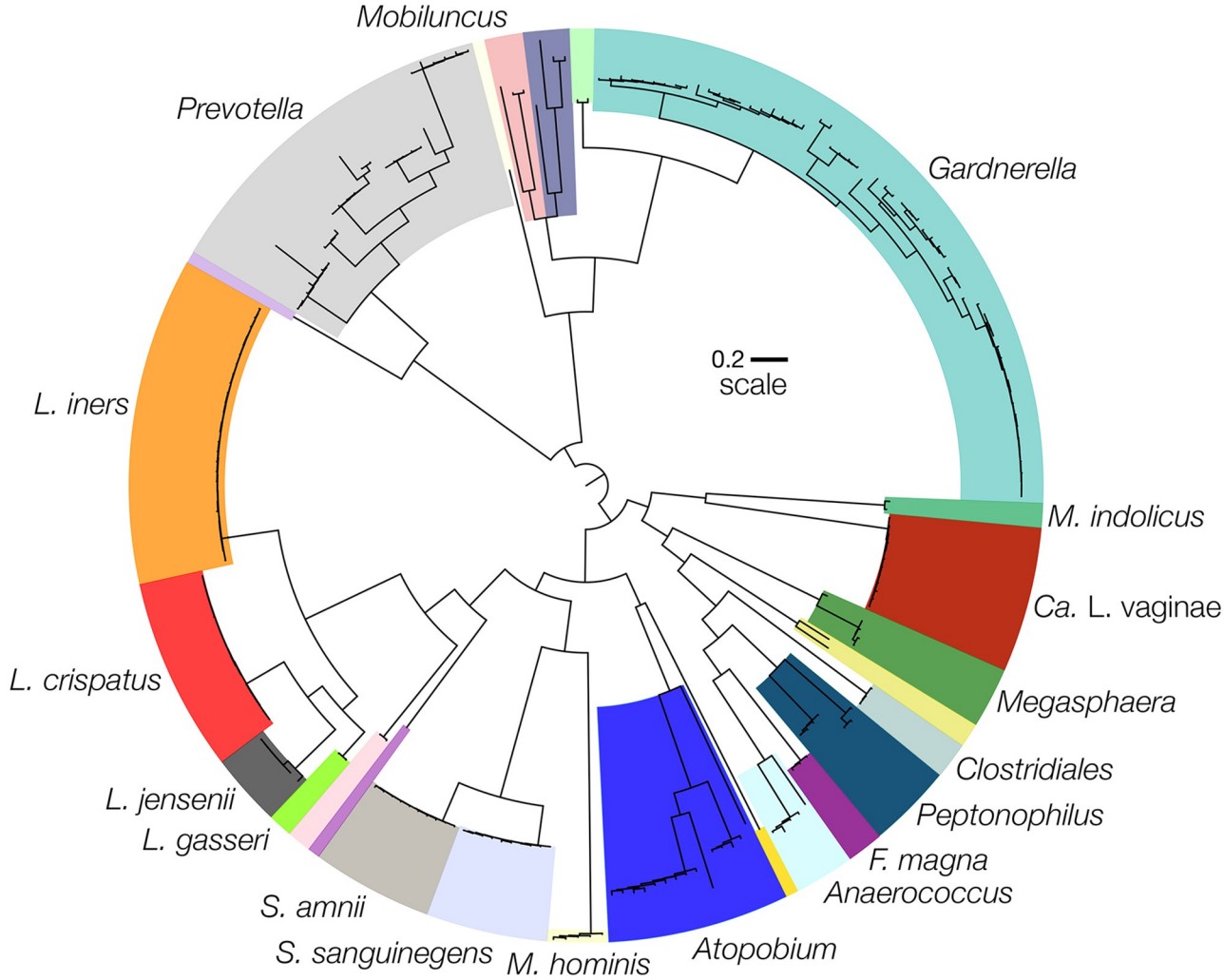

**Fig 3. Phylogenetic tree displaying the taxonomic diversity of the 225 metagenome assembled genomes (MAGs) derived from the shotgun metagenomic data generated for 22 families.** The maximum likelihood phylogeny was established using a concatenated alignment of the amino acid sequences for 13 orthologous genes which were found to be present in at least 99% of the MAGs: *gapD, gltX, ileS, pheS, pheT, cysS, hisS, uvrD, ruvX, rpsO, Ffh, obgE*, and *lepA*.

*Prevotella*, 10 *S. amnii*, and 10 *S. sanguinegens*. The remaining 48 MAGs comprised 24 species. We were able to recover four *Gardnerella* MAGs from a single metagenome which contained *Gardnerella*.

## Identification of shared strains

To identify pairs of MAGs which originate from the same bacterial strain we used the inStrain tool and associated workflow. If the vaginal microbiota is vertically transmitted, we expected that mothers and daughters which had species in common, might also have strains in common. Furthermore, if this transmission had happened at the time of birth, the mother and daughter strains should also not be identical but instead show some degree of sequence divergence consistent with the amount of time past. For this reason, we used a stringent threshold for defining shared strains of at least 99.9% sequence identity and at least 70% overlap. Among our set of 225 MAGs, we identified 49 pairs which met this threshold. Among these, ten were

**Table 1. MAGs identified in mother-daughter pairs.**

| Taxonomy | Family | Daughter's age | Rel. Abund. Daughter† | Mother's age | Rel. Abund. Mother† | Substitution Rate* |
|---|---|---|---|---|---|---|
| "*Ca.* L. vaginae" | 21 | 18 | 9.21% | 41 | 2.80% | 1.48E-05 |
| *A. vaginae* | 30 | 14 | 2.22% | 42 | 0.46% | 1.05E-06 |
| Clostridiales Family | 5 | 14 | 2.15% | 36 | 2.31% | 5.18E-06 |
| *Gardnerella* | 30 | 14 | 47.42% | 42 | 16.56% | 1.98E-05 |
| *L. crispatus* | 3 | 15 | 44.54% | 32 | 72.13% | 4.00E-06 |
| *L. crispatus* | 1 | 15 | 80.91% | | 75.57% | 9.48E-06 |
| *L. crispatus* | 5 | 14 | 31.88% | 36 | 39.26% | 6.26E-06 |
| *L. iners* | 30 | 14 | 12.34% | 42 | 5.36% | 8.32E-06 |
| *P. timonensis* | 41 | 15 | 16.16% | 47 | 1.63% | 7.50E-06 |
| *P. timonensis* | 5 | 14 | 2.18% | 36 | 2.83% | 3.24E-05 |

*per base pair per year.

†Relative abundance in community.

between a mother and daughter from the same family, representing six mother-daughter pairs. These ten pairs of strains were found at both high and low relative abundances and their abundance was generally similar between the mother's and daughter's communities (Table 1). Daughters in pairings which were found to share strains were younger at the time of sampling than those in pairings which were not found to share strains (15.17 versus 19.62; t = -4.31; p<0.001). No trend was observed with the mother's age (39.6 versus 41.9; t = -0.81; p = 0.45). Of these six mother-daughter pairs, three were found to share *L. crispatus* strains (families 1, 3, and 5) with family 5 also sharing MAGs classified as Clostridiales and *P. timonensis*. Family 30 shared MAGs classified *Gardnerella*, *L. iners*, and *A. rimae* while family 21 shared only "*Ca.* L. vaginae" and family 41, only *P. timonensis*. The remaining 39 instances of shared strains were between daughters and mothers from different families (n = 18), mothers from different families (n = 13), and daughters from different families (n = 8). These pairings were parsed into a diagram representing the network of shared strains among the mothers and daughters in this study (Fig 4). A large part of this network was comprised of a *L. crispatus* and a *L. jensenii* strain which were identified in five and four metagenomes, respectively.

## Comparison of sequence identity among strain shared within versus between families

We next asked whether the strains identified as shared within mother-daughter pairs were more or less similar than those shared between families. Sequence similarity was measured as the number of singe nucleotide polymorphisms (SNPs) per megabase pair of aligned sequence. We found strains shared within families trended to being more similar to one another than those shared between families, but this difference was not significant (Fig 5, W = 136, p = 0.149). For the strains shared within families these values were used in combination with the daughter's age to calculate the per year substitution rate under the hypothesis of vertical transmission at the time of birth. These values ranged from $1.05 \times 10^{-6}$ to $3.24 \times 10^{-5}$ per base pair per year and are listed in Table 1.

## Discussion

The origin of the bacterial strains which constitute the vaginal microbiota is not currently known. We found limited evidence for vertical transmission of these bacteria from a mother to her daughter which had persisted through the daughter's adolescence. While some mother-

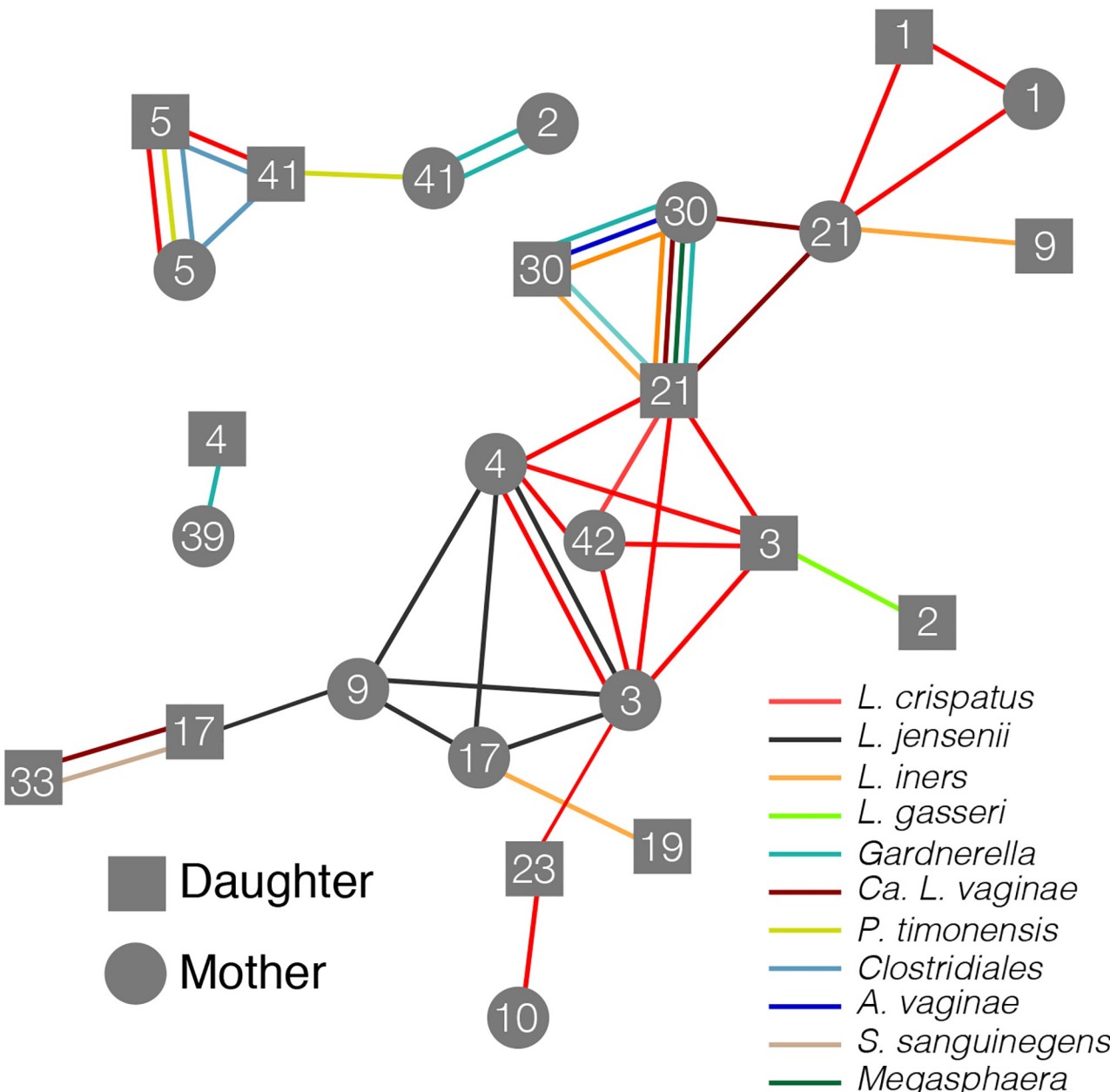

**Fig 4. Network diagram of shared bacterial strains identified in this cohort.** A stringent threshold, 99.9% sequence identity, 70% coverage, was used to identify shared strains. Lines represent the shared bacterial strains and connect the participants in which the strain was found. Mothers are represented by circles and daughters by squares. Numbers on the nodes signify the family and can be linked back to the taxonomic profile of the participant using Fig 1.

daughter pairs were found to have communities of similar taxonomic composition, the observed similarities could be explained by chance. A small subset of the mother-daughter pairs were also found to have bacterial strains in common, consistent with vertical transmission, but shared strains were more frequently identified in unrelated individuals. These results do not eliminate vertical transmission as a possible mechanism by which the vaginal microbiota is founded but rather suggest that mothers and daughters do not necessarily have similar vaginal microbiota, later in life. Because we examined the communities years after the birth of

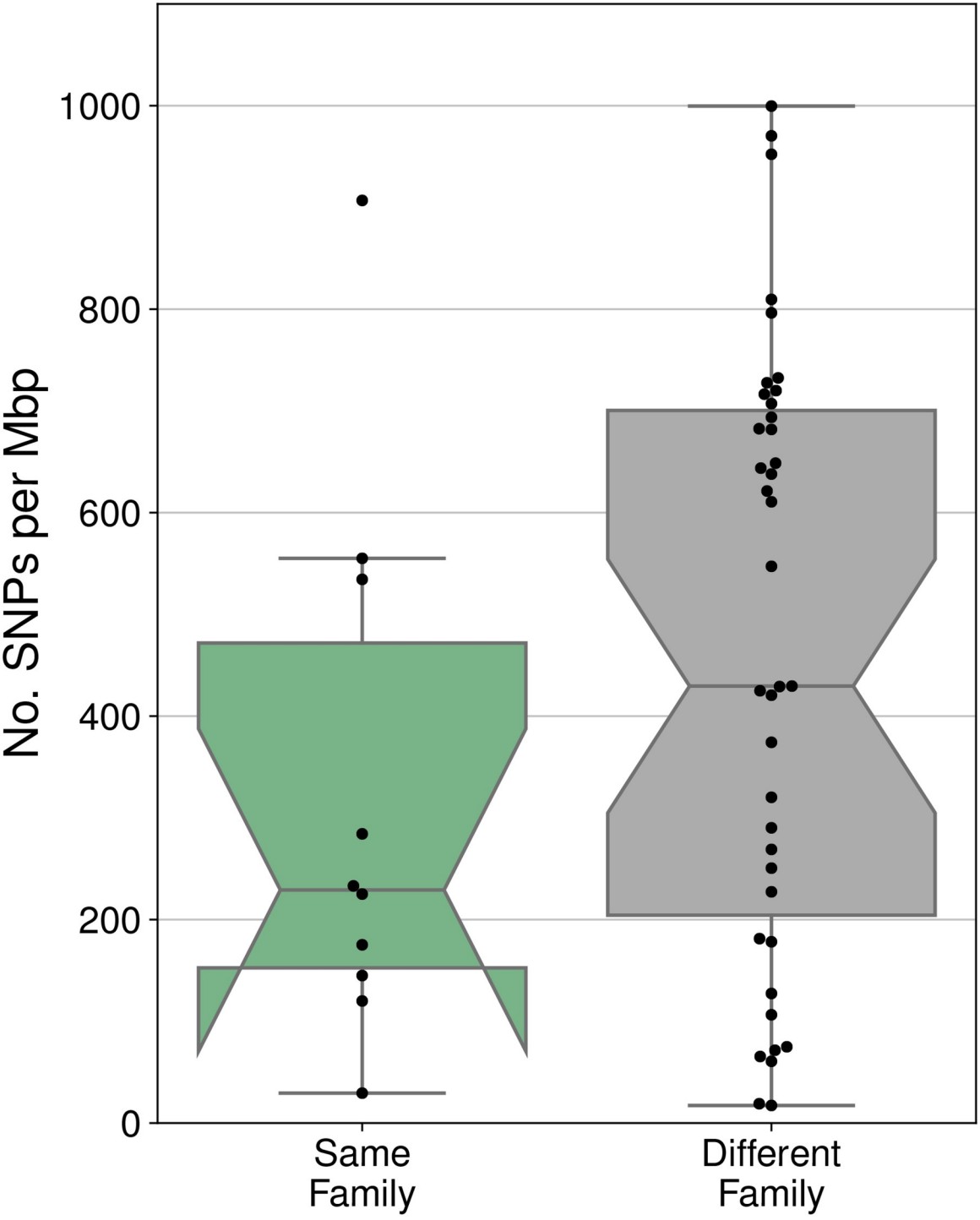

**Fig 5. Number of single nucleotide polymorphisms identified in strains found to be shared between members the same family or between members of different families.** Values are scaled per mega base pair of compared sequence.

the daughter, there was plenty of time for either the mother's or the daughter's vaginal microbiota to experience strain turnover. Longitudinal studies have indicated that the vaginal microbiota of reproductive-age women does experience changes in composition over time, although the studies followed women for only a few months [65,66]. Daughters that were found to share strains with their mother were also younger than those which were not, further suggesting that time may play a role. It is not difficult to imagine that the populations of some bacterial strains might go extinct over the course of a person's life. The mechanisms by which new bacterial strains might be introduced into the vaginal microbiota are not well understood. Unprotected vaginal sex and other sexual practices could result in the introduction of new bacterial strains [67–69], but there are likely other mechanisms as well. Strain turnover in either the mother or the daughter's vaginal microbiota would erode any signal of vertical transmission.

In our analysis, six mother-daughter pairs were found to have matching bacterial strains in their vaginal microbiota. Under the vertical transmission at the time of birth, substitutions are expected to accumulate in both the mother's and the daughter's populations, as they evolve independently post-transmission. We used the daughter's age to calculate the substitution rate under this hypothesis and arrived at values between $3^*10^{-5}$ and $1^*10^{-6}$ substitutions per site-year for each shared strain. It is difficult to say how if our observed values fit the vertical transmission narrative, as the expected substitution rate is not well understood. A study on the population genomics of *Neisseria gonorrhoeae*, a sexually transmitted pathogen, estimated a rate of $3^*10^{-5}$ substitutions per site-year [70]. Another study surveyed of the substitution rate experienced by a number of human pathogens estimated rates between $10^{-5}$ and $10^{-8}$ substitutions per site-year [71]. They also found a strong negative relationship between the estimated substitution relationship and the timescale over it was measured. The authors suggest that this relationship results from the accumulation of deleterious mutations which have yet to be purged by purifying selection [71]. The timescale separating our hypothesized vertically transmitted strains is rather short, consistent with our relatively high estimated substitution rates. Many of the bacteria common to the human vagina have reduced genome sizes and have lost components of DNA repair machinery [72,73]. These bacteria may experience higher than average mutation rates [74–77] which could lend itself to higher estimates of their substitution rate [78]. We cannot say for certain that our observation of shared strains among these six mother-daughter pairs is the result of the vertical transmission at time of birth, but we find this explanation is reasonable.

The majority of shared strains were identified in women from unrelated families. We hypothesize that this observation may reflect the biogeography of vaginal bacteria. The mothers and daughters included in this study were all living in the Baltimore metropolitan area at the time of sampling. These individuals may be more likely to share bacterial strains with one another simply due to their geographic proximity. Lourens Baas Becking put forth the hypothesis that bacteria did not display biogeographic patterns, suggesting that in the microbial world "everything is everywhere, but the environment selects" [79]. In the years since Bass-Becking put forth his hypothesis, there have been several demonstrations to the contrary [80–84]. The dispersal of some bacterial species appears to be constrained leading them to exhibit biogeographic patterns. We suggest that many of the species common to the vaginal microbiota (e.g. *L. crispatus*, *L. iners*, *L. jensenii*, *G. vaginalis*, "Ca. L. vaginae", *A. vaginae*) are among those likely to have their dispersal constrained. With the except of *L. crispatus*, which is sometimes found in the intestinal tracts of chickens [85,86], these species are not routinely found anywhere other than the vagina. Many of these bacteria are also fastidious and require anaerobic conditions for their robust growth. It is therefore not clear how they might disperse over the great distances necessary to erode biogeographic patterns, except by means of their host. If dispersal is primarily achieved via sexual activity, sexual networks could underpin biogeographic patterns observed for vaginal bacteria [87]. Additional studies of participants from around the world

might further illuminate these biogeographic patterns and the factors which govern them. Results from such studies could have translational impact and would be informative on the necessity of developing geo-adapted probiotic formulations to modulate the vaginal microbiota.

We implemented a stringent sequence similarity cutoff ($\geq$99.9% similarity) to identify strains which may have been vertically transferred from mother to daughter. This is because it is not enough to identify instances where two bacterial assemblies belong to the same lineage. Their genome sequences must also be similar enough that any observed sequence differences can be explained by the post-transmission evolution of the two populations. In our case, the transmission was hypothesized to have occurred at birth, meaning that the two populations had evolved independently, in most cases, for about two decades. Other studies which have examined the maternal transmission of microbes from mother to offspring have done so shortly after birth [24–26]. In this case, there should be minimal sequence differences between the two populations of vertically transferred strains. It makes sense then, to utilize even more stringent sequence similarity cutoffs than that used here (e.g. $\geq$99.99% similarity). Marker gene based tools like StrainPhlan [23], do not have the capability to implement such genome-wide sequence similarity cutoffs and therefore may not be the best tool for this analysis. We, as have others [63], advocate for the use of appropriate sequence similarity cutoffs to identify recent microbial transfer events. Identifying the same species or even the same bacterial strain in two samples is not enough.

This study has a number of important limitations which should be considered. First, we examined the mother's and daughter's vaginal microbiota not at birth, but instead sometime after the daughter had experienced menarche. This means that there was plenty of time for the daughter or the mother to gain or lose bacterial strains in their communities. Second, we are missing a great deal of metadata which could help explain why some mother-daughter pairs were found to share bacterial strains, and some were not. For example, we do not know which daughters were born by cesarean section and which were vaginally delivered. Nor do we have sexual behavior or partner history data for the participants in this study. Third, the participants had originally been enrolled in a douching intervention study: douching may influence the composition of the vaginal microbiota [88]. Finally, the cohort examined was relatively small and included only women in the greater Baltimore metropolitan area who identified as Black or African American.

Yet even with these limitations, we did identify several mother-daughter pairs which did share strains, which were similar in sequence enough to have been vertically transmitted at the time of birth. These results motivate future studies which investigate the extent to which the vaginal microbiota is vertically transmitted and the importance of this event to the daughter's future reproductive health. We also identified shared strains in unrelated individuals, suggesting that vaginal bacteria might display biogeographic patterns. These patterns could be confirmed by large scale, multi-regional studies which examine the extent to which strains of vaginal bacteria show geographic specificity.

## Supporting information

**S1 Table. Taxonomic compositions.** Table containing the taxonomic composition derived from the 16S rRNA gene amplicon sequencing.
(CSV)

**S2 Table. Metagenome assembled genomes inventory.** Table describing the completeness and degree of contamination of the metagenome assembled genomes generated in this study.
(CSV)

## Author Contributions

**Conceptualization:** Michael T. France, Anne M. Rompalo, Rebecca M. Brotman, Jacques Ravel.

**Data curation:** Michael T. France, Sarah E. Brown.

**Formal analysis:** Michael T. France.

**Funding acquisition:** Anne M. Rompalo, Rebecca M. Brotman, Jacques Ravel.

**Investigation:** Michael T. France, Sarah E. Brown.

**Methodology:** Michael T. France.

**Project administration:** Jacques Ravel.

**Software:** Michael T. France.

**Supervision:** Anne M. Rompalo, Rebecca M. Brotman, Jacques Ravel.

**Visualization:** Michael T. France.

**Writing – original draft:** Michael T. France.

**Writing – review & editing:** Michael T. France, Sarah E. Brown, Anne M. Rompalo, Rebecca M. Brotman, Jacques Ravel.

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
