## [Decision Letter · Decision Letter 0]

8 Sep 2022

PONE-D-22-16695Identification of shared bacterial strains in the vaginal microbiota of related and unrelated reproductive-age mothers and daughters using genome-resolved metagenomicsPLOS ONE

Dear Dr. Ravel,

Thank you for submitting your manuscript to PLOS ONE. After careful consideration, we feel that it has merit but does not fully meet PLOS ONE’s publication criteria as it currently stands. Therefore, we invite you to submit a revised version of the manuscript that addresses the points raised during the review process.

We look forward to receiving your revised manuscript.

Kind regards,

Andrew B. Onderdonk

Academic Editor

PLOS ONE

Journal Requirements:

“Dr. Ravel is a co-founder of LUCA Biologics, a biotechnology company focused on translating microbiome research into live biotherapeutics for women’s health. All other authors declare that they have no competing interests.”

Additional Editor Comments:

I apologize for the length of time evaluation of this manuscript has taken, but I had an exceptionally difficult time finding competent reviewers for this manuscript. I agree with the comments of the reviewer.

Reviewers' comments:

Reviewer's Responses to Questions

**Comments to the Author**

1. Is the manuscript technically sound, and do the data support the conclusions?

Reviewer #1: Yes

2. Has the statistical analysis been performed appropriately and rigorously? 

Reviewer #1: Yes

3. Have the authors made all data underlying the findings in their manuscript fully available?

Reviewer #1: Yes

4. Is the manuscript presented in an intelligible fashion and written in standard English?

Reviewer #1: Yes

5. Review Comments to the Author

Reviewer #1: This work aims to identify shared bacterial strains in vaginal

microbiota of reproductive-age daughters and mothers (ages 14-27,

32-51; n=39 pairs) from a metropolitan area to evaluate the hypothesis

that vertical transmission is present in vaginal microbiota. Overall

similarity in microbiota composition was evaluated for all volunteers

with metataxonomics, and additional genome-resolved metagenomics was

used to detect shared strains in a subset of families (n=22). In

general, metataxonomic similarities did not exceed random expectations

and the strain-level metagenomic approach indicated that less strains

were shared between family members (this correlated with age; younger

daughters shared more strains with their mothers) than unrelated

pairs, one explanation being that biogeography might be a stronger

driver for similarities in the vaginal microbiota of reproductive-age

women than family relations, although some evidence for vertical

transmission was present.

I focus on bioinformatics aspects of the work in the evaluation.

Overall, the methodology is sound and clearly reported, and the

conclusions are supported by the presented material. Standard methods

for bioinformatics and data analysis have been used; it is interesting

to note the use of the less commonly used Yue-Clayton similarity index

in the analysis. Limitations of the study are discussed adequately,

and relevant literature is cited.

I commend the authors for not only making the data and code available

but doing this in a well organized format including an open license

(GPL-3).

I only have the following minor comments on this report.

# Minor

- Fig 1: might be useful to check if there are alternative ways to

visualize the M/D relations in this figure; the comparison is

complicated by the large number of colors (species) and limited

overlap in M/D pairs. Perhaps showing the results per species &

group level variations across all subjects (e.g. difference in the

abundance between M/D?) could help to highlight the most

distinguished similarity patterns. Just a suggestion to consider.

- Consider using a more permissive and broadly compatible open source

license, such as MIT, BSD-2-clause, or Artistic 2 (for discussion,

see e.g. Morin et al. 2012

https://journals.plos.org/ploscompbiol/article?id=10.1371/journal.pcbi.1002598). Also

consider that many open source developers (including Linux kernel)

have chosen to stick to GPL2.

- A README file in the source code could help to navigate and

interpret the material

- line 182; was the Student's t test assumptions fulfilled

(approximately normally distributed groups), or would a

non-parametric test (Wilcoxon) be justified?

6. PLOS authors have the option to publish the peer review history of their article (what does this mean?). If published, this will include your full peer review and any attached files.

Reviewer #1: **Yes: **Leo Lahti

---

## [Author Response · Author response to Decision Letter 0]

23 Sep 2022

Response to reviewer’s comments

Responses to comments in blue

Reviewer #1: This work aims to identify shared bacterial strains in vaginal microbiota of reproductive-age daughters and mothers (ages 14-27, 32-51; n=39 pairs) from a metropolitan area to evaluate the hypothesis that vertical transmission is present in vaginal microbiota. Overall similarity in microbiota composition was evaluated for all volunteers with metataxonomics, and additional genome-resolved metagenomics was used to detect shared strains in a subset of families (n=22). In general, metataxonomic similarities did not exceed random expectations and the strain-level metagenomic approach indicated that less strains were shared between family members (this correlated with age; younger daughters shared more strains with their mothers) than unrelated pairs, one explanation being that biogeography might be a stronger driver for similarities in the vaginal microbiota of reproductive-age women than family relations, although some evidence for vertical transmission was present.

I focus on bioinformatics aspects of the work in the evaluation.

Overall, the methodology is sound and clearly reported, and the conclusions are supported by the presented material. Standard methods for bioinformatics and data analysis have been used; it is interesting to note the use of the less commonly used Yue-Clayton similarity index in the analysis. Limitations of the study are discussed adequately, and relevant literature is cited.

I commend the authors for not only making the data and code available but doing this in a well-organized format including an open license (GPL-3).

I only have the following minor comments on this report.

We thank the reviewer for their time and effort in evaluating our work.

# Minor

- Fig 1: might be useful to check if there are alternative ways to visualize the M/D relations in this figure; the comparison is complicated by the large number of colors (species) and limited overlap in M/D pairs. Perhaps showing the results per species & group level variations across all subjects (e.g. difference in the abundance between M/D?) could help to highlight the most

distinguished similarity patterns. Just a suggestion to consider.

We explored additional options for how best to display Figure 1 but ultimately chose to keep the current version. A motivating reason for our decision was that the current plot allows us to display the family’s community composition and label them by number. These numbers are re-used in the network diagram and allow the reader to refer back to the family’s community composition.

- Consider using a more permissive and broadly compatible open source

license, such as MIT, BSD-2-clause, or Artistic 2 (for discussion,

see e.g. Morin et al. 2012

https://journals.plos.org/ploscompbiol/article?id=10.1371/journal.pcbi.1002598). Also

consider that many open source developers (including Linux kernel)

have chosen to stick to GPL2.

We have revised the open-source license to the more permissive MIT, as suggested by the reviewer. 

- A README file in the source code could help to navigate and

interpret the material

A README file was added to the Github page to help navigate the provided material.

- line 182; was the Student's t test assumptions fulfilled (approximately normally distributed groups), or would a non-parametric test (Wilcoxon) be justified?

Upon further examination, the assumption of normality was not met. For this reason, we have switched to the non-parametric test as suggested by the reviewer. Note the p-value did not change.

L174-176 “Sequence similarity between shared strains identified in the same family versus different families were compared using a Wilcoxon rank sum test.”

L242-244 “We found strains shared within families trended to being more similar to one another than those shared between families, but this difference was not significant (Figure 5, W15.1=-136, p=0.149).”

---

## [Editor Report · Decision Letter 1]

26 Sep 2022

Identification of shared bacterial strains in the vaginal microbiota of related and unrelated reproductive-age mothers and daughters using genome-resolved metagenomics

PONE-D-22-16695R1

Dear Dr. Ravel,

We’re pleased to inform you that your manuscript has been judged scientifically suitable for publication and will be formally accepted for publication once it meets all outstanding technical requirements.

Kind regards,

Andrew B. Onderdonk

Academic Editor

PLOS ONE
---

## [Editor Report · Acceptance letter]

29 Sep 2022

PONE-D-22-16695R1 

Identification of shared bacterial strains in the vaginal microbiota of related and unrelated reproductive-age mothers and daughters using genome-resolved metagenomics 

Dear Dr. Ravel:

I'm pleased to inform you that your manuscript has been deemed suitable for publication in PLOS ONE. Congratulations! Your manuscript is now with our production department. 

Kind regards, 

on behalf of

Dr. Andrew B. Onderdonk 

Academic Editor

PLOS ONE